# Promotive Role of 5-Aminolevulinic Acid or Salicylic Acid Combined with Citric Acid on Sunflower Growth by Regulating Manganese Absorption

**DOI:** 10.3390/antiox12030580

**Published:** 2023-02-25

**Authors:** Juanjuan Li, Jianmin Pan, Ullah Najeeb, Hossam S. El-Beltagi, Qian Huang, Huaijian Lu, Ling Xu, Bixian Shi, Weijun Zhou

**Affiliations:** 1Key Laboratory of Plant Secondary Metabolism and Regulation of Zhejiang Province, College of Life Sciences and Medicine, Zhejiang Sci-Tech University, Hangzhou 310018, China; 2Institute of Crop Science, Ministry of Agriculture and Rural Affairs Laboratory of Spectroscopy Sensing, Zhejiang University, Hangzhou 310058, China; 3Queensland Alliance for Agriculture and Food Innovation, Centre for Plant Science, The University of Queensland, Toowoomba, QLD 4350, Australia; 4Agricultural Research Station, Office of VP for Research & Graduate Studies, Qatar University, Doha 2713, Qatar; 5Agricultural Biotechnology Department, College of Agriculture and Food Sciences, King Faisal University, Al-Ahsa 31982, Saudi Arabia; 6Biochemistry Department, Faculty of Agriculture, Cairo University, Giza 12613, Egypt; 7Institute of Economic Crops, Xinjiang Academy of Agricultural Sciences, Urumqi 830091, China

**Keywords:** 5-aminolevulinic acid, salicylic acid, citric acid, Mn absorption, antioxidant defense system

## Abstract

Manganese (Mn) is an essential nutrient in most organisms. Establishing an effective regulatory system of Mn absorption is important for sustainable crop development. In this study, we selected sunflower as the model plant to explore the effects of 5-aminolevulinic acid (ALA) or salicylic acid (SA) combined with citric acid (CA) on Mn absorption. Six-leaf-old sunflower plants were exposed to 0.8 g kg^−1^ Mn for one week and then treated with chelating agents, i.e., CA (10 mmol kg^−1^), and different concentrations of ALA and SA for one week. The results showed that Mn-treated plants had significantly increased H_2_O_2_, O_2_^−^ and MDA contents in leaves compared with the control. Under the Mn + CA treatment, ALA or SA2 significantly activated the antioxidant defense system by increasing SOD, POD and CAT activities in leaves. Moreover, the application of CA significantly increased the Mn uptake in sunflower roots compared with Mn treatment alone; however, did not accelerate the translocation efficiency of Mn from sunflower roots to shoots. Moreover, ultrastructural and RT-qPCR results further demonstrated that ALA/SA could recover the adverse impact of excessive Mn accumulation in sunflowers. Like a pump, ALA/SA regulated the translocation efficiency and promoted the transportation of Mn from roots to shoots. This study provides insights into the promotive role of ALA/SA combined with CA on sunflower growth by regulating Mn absorption, which would be beneficial for regulating Mn absorption in soil with an Mn deficit.

## 1. Introduction

Manganese (Mn) is an essential micronutrient for plants. Being part of the oxygen-evolving complex in photosystem II (PS II), Mn plays an indispensable role in photosynthesis [1] and plant development [2]. It also acts as a cofactor or activator for various enzymes such as manganese superoxide dismutase (Mn-SOD), RNA polymerases, decarboxylases, etc. [3]. Mn bioavailability in the soil is regulated by the soil pH level, water content and porosity [4], and its elevated levels can inhibit plant growth. In many agricultural systems, Mn availability has increased to toxic levels for plant growth due to inappropriate fertilization [5]. For example, Mn toxicity has been reported in rice [6], soybean and sunflower [5]. Mn toxicity in plants is linked with its negative impact on key cellular organelles such as the chloroplast and nucleus [7]. Therefore, application of suitable concentration of Mn is vital for plant growth and development.

Exploring a strategy of applying a plant growth regulator (PGR) and/or a chelator could be useful to regulate plant Mn absorption in Mn-deficient soil. Phytoextraction could be applied to recover excessive Mn in the soil, due to its environmentally friendly nature [8]. One kind of PGR, 5-aminolevulinic acid (ALA), has been reported to promote cadmium accumulation in sunflower plants [9] and increase the tolerance of *Brassica napus* under Cd stress [10,11] or lead toxicity [12]. Another PGR, salicylic acid (SA), is a phenolic compound involved in the improvement of plant tolerance to various abiotic stresses such as the herbicide quinclorac [13,14], the heavy metal cadmium [15], salinity [16], drought [17], etc. Thus, SA could act as a potential tool in mitigating the adverse effects of abiotic stress in plants [18]. Chelators such as citric acid (CA) and ethylene diamine tetra acetic acid (EDTA)-induced phytoextraction have been used to enhance the uptake and translocation of metals from soil [7]. Nevertheless, EDTA is unsuitable for practical application in agricultural systems due to its slow degradation [19]. Comparatively, low-molecular-weight organic acids (LMWOA) such as CA are biodegradable and effective chelators for promoting phytoremediation of metal-polluted soils [20].

Sunflower (*Helianthus annuus* L.) is an important oilseed and industrial crop cultivated across many parts of the world [21]. Regulation of Mn absorption could be crucial for sunflower production. Earlier studies reported that sunflower has become the ideal crop to phytoremediate heavy-metal-polluted soils due to its morphological traits such as fast growth, large biomass and deep roots. Phytoextraction with *H. annuus* has primarily focused on heavy metals such as Cd, lead (Pb) and uranium (U) [22,23,24,25,26]. To date, the regulation of Mn absorption in *H. annuus* has not been fully studied. In this study, we explored the synergistic effects of CA and ALA/SA on Mn absorption in *H. annuus*, which could provide an important strategy for regulating plant Mn absorption.

## 2. Experimental Section

### 2.1. Plant Materials and Growth Conditions

Sunflower (*Helianthus annuus* L.) seeds (cultivar TP3316) were obtained from the Institute of Plant Protection, Inner Mongolia Academy of Agricultural and Animal Husbandry Sciences, Hohhot, China. The seeds were germinated on a petri dish in dark for two days, and then sown in culture medium (perlite: vermiculite: nutrient soil = 1:2:3) in plastic pots (130 mm × 110 mm). The plants were irrigated with non-ionized water for two weeks. Six-leaf uniform-size seedlings were treated with Mn (0.80 g kg^−1^, MnSO_4_). CA (20 mmol kg^−1^) treatment was conducted one week after Mn application. Four-week-old plants (one week after CA treatment) were then sprayed either with water or with different concentrations of ALA (10, 20 mg L^−1^) or SA (50, 100 mg L^−1^) once droplets formed. Fifteen treatments in total are presented in Appendix A. All the treatment concentrations of Mn, CA, ALA and SA were selected based on our preliminary experiments. All the plants were grown under 200 μmol m^−2^ s^−1^ active photon flux density, 24/20 °C (day/night) temperature, 60–70% relative humidity and 14/10 h (light/dark) photoperiod.

### 2.2. Plant Biomass, Chlorophyll Content, Malondialdehyde (MDA) and Reactive Oxygen Species (ROS) Analysis

Five weeks old plants were removed from the pots, separated into shoots and roots, and rinsed carefully with deionized water, then sucked dry for fresh weight. The shoots and roots were dried at 75 ± 5 °C until constant for dry weight. Further, 0.1 g fresh leaves without veins were homogenized and extracted in a 5 mL mixture containing 45% ethanol, 45% acetone and 10% distilled water in the dark. The supernatant of each sample was used to determine chlorophyll contents (chlorophyll a, b and total chlorophyll) at 663 nm and 645 nm and carotenoid contents at 470 nm according to the methods of Porra et al. [27] with some modifications. The MDA and ROS including hydrogen peroxide (H_2_O_2_), superoxide radical (O_2_^−^) and extra-cellular hydroxyl radicals (^−^OH) of fresh leaves and roots were determined by following the protocols of Xu et al. [9].

### 2.3. Antioxidant and Non-Antioxidant Enzyme Activities

Antioxidant enzyme activities were determined using fresh plant tissues (0.3 g of root and leaf each). These tissues were homogenized in 50 mM potassium phosphate buffer (pH 7.8) at 4 °C and centrifuged for 20 min at 10,000 rpm. The supernatant was collected and used for assaying different antioxidant enzymes. The photochemical nitro blue tetrazolium (NBT) protocol was followed for total superoxide dismutase (SOD) determination [28]. Zhou and Leul’s [29] method was used for measuring peroxidase (POD) activity. Catalase (CAT) activity was determined by calculating the degradation of H_2_O_2_ (extinction coefficient 39.4 mM^−1^ cm^−1^) for 60 s at 240 nm [30]. For ascorbate peroxidase (APX) measurement, an extinction coefficient of 2.8 mM^−1^ cm^−1^ was used. APX activity was measured at 290 nm for 30 s after adding H_2_O_2_ into the reaction mixture [31].

The glutathione content was investigated according to the method of Law [32] with some modifications. Root and leaf tissues (0.1 g) were homogenized with 1 mL of 3% TCA (containing 5 mM EDTA) and centrifuged at 15,000 rpm for 15 min. The total glutathione was determined in a reaction mixture containing 140 μL of 0.3 mM NADPH, 20 μL of 6 mM DTNB, 10 μL of glutathione reductase (10 units mL^−1^) and 30 μL supernatant at 412 nm (extinction coefficient 6.2 mM^−1^ cm^−1^). Oxidized glutathione (GSSG) was determined by subtracting GSH from the total glutathione content.

### 2.4. Heavy Metal Analysis

To detect the elemental concentrations, dried plant samples (0.05 g) were digested with 8 mL concentrated nitric acid (guaranteed reagent, GR) in closed Teflon vessels using a microwave digestion instrument. The solution was heated until nearly dry and then washed with ddH_2_O. An innovative single-quadrupole iCAP^TM^ RQ ICP-MS (Thermo Fisher, USA) was used for analyzing Mn concentrations in shoot and root tissues. We also detected the concentrations of other main metallic (Mn, Na, Mg, Al, P, K, Ca, Fe, Cu and Zn) and non-metallic (B) elements in sunflower shoots and roots for further analyzing the correlation between the concentrations of Mn and these elements. Mn uptake, translocation and movement in plant tissues were calculated using a previously established Xu et al. [9] protocol.

### 2.5. Transmission Electron Microscopy

Fresh leaf and root-tip samples from treated plants were used for microscopic observations. The samples were dehydrated and embedded in a suitable resin using a Li et al. [33] protocol. A transmission electron microscope H-7650 (Hitachi, Ibaraki, Japan) was used for analyzing the images.

### 2.6. Gene Expression Analysis

Total RNA was extracted from 0.1 g frozen leaf and root tissues of sunflower using RNAprep Pure Plant Kit (TIANGEN, Beijing, China) following the manufacturer’s procedure. RNA integrity was analyzed on a 1.0% agarose gel. The RNA quantity was determined using a NanoDrop 2000 Spectrophotometer (Thermo Fisher Scientific, Massachusetts, USA) [34,35]. TaKaRa PrimeScript^TM^ RT Master Mix (Perfect for Real Time) was used for cDNA synthesis. All cDNA samples were assayed in the QuantStudio 6 Flex Real-Time PCR System (Thermo Fisher, Marsiling, Singapore) using TB Green^TM^
*Premix Ex* Taq^TM^ II (Tli RNaseH Plus) (TaKaRa) with the following procedures: 95 °C for 30 s, followed by 40 cycles of denaturation at 95 °C for 5 s, annealing and extension at 58 °C for 30 s according to Li et al. [36].

Based on the physiological data of different treatments, we selected the effective concentration of PGRs for real-time quantitative PCR (RT-qPCR) experiments, i.e., Mn, Mn + ALA2, Mn + SA2 or Mn + CA, Mn + CA + ALA2 and Mn + CA + SA2. In this study, we examined the expression of 20 genes related to stress tolerance in treated sunflower roots and leaves with three biological replicates and three technical replicates. Primers used for RT-qPCR are listed in Appendix A. *EF-1α* was used as a reference gene.

### 2.7. Statistical Analysis

All data presented in this study were analyzed with Excel 2010, SPSS 23.0, TBtools v1.0987 and Graphpad Prism 8.4.2. Meanwhile, all the experiments were conducted with three biological replicates and the values described in the results section are the means of three replicates ± standard error (SE). Two-way analysis of variance (ANOVA) was applied followed by Tukey’s multiple comparisons test between the means of treatments to determine significant differences (* represents *p*  <  0.05; **, *** and **** represent *p*  <  0.01,  0.001 and 0.0001, respectively; and ns indicates no significant difference).

## 3. Results

### 3.1. Plant Growth, Leaf Chlorophyll and Carotenoid Contents

Mn-treated sunflower plants were significantly taller and heavier (shoot dry and fresh weight) compared with control plants (Table 1). The chelator CA (20 mmol kg^−1^) further significantly increased shoot and root biomass and plant height of Mn-treated plants, and reached the highest level among all the treatments. In the presence of Mn, a lower concentration ALA (ALA1) or SA (SA1) significantly promoted shoot fresh weight (26.29%, 22.36%, respectively), shoot dry weight (14.01%, 10.00%, respectively), and plant height (7.02%, 5.40%, respectively) compared with Mn treatment alone. However, ALA and SA significantly decreased shoot and root weight and plant height of Mn + CA-treated plants.

In contrast to plant growth, Mn-treated plants had significantly lower leaf chlorophyll a, b and total chlorophyll contents than the control (Figure 1). Further significant reductions in chlorophyll a, b and total chlorophyll contents were recorded in Mn + CA-treated sunflower leaves (Figure 1). However, the carotenoid content in Mn-treated sunflower leaves was significantly higher (20.48%) than that of the control, and this was further enhanced after further application of CA. Interestingly, the effects of ALA or SA on the carotenoid content were obvious in Mn-treated sunflower plants, except ALA1. However, ALA or SA application significantly decreased carotenoid content in Mn + CA treated sunflower leaves, except ALA2. In the absence of Mn, 20 mg L^−1^ ALA (ALA2) significantly increased chlorophyll a and carotenoid content as compared with the control. ALA2, SA1 and SA2 significantly increased chlorophyll a, chlorophyll b and the total chlorophyll contents of sunflowers under the Mn + CA treatment. For example, ALA2, SA1 and SA2 treatment increased the total chlorophyll content in leaves of Mn + CA-treated plants by 28.61%, 77.03% and 83.68%, respectively (Figure 1).

### 3.2. Reactive Oxygen Species (ROS) and Lipid Peroxidation (MDA)

Lipid peroxidation in the leaves and roots was evaluated with ROS and MDA contents. Mn-treated plants had significantly elevated H_2_O_2_, O_2_^−^ and MDA levels in root and leaf tissues compared with the control (Table 2). For instance, Mn treatment increased H_2_O_2_, O_2_^−^ and MDA contents of leaves by 41.62%, 128.12% and 13.16%, and those of roots by 17.08%, 16.62% and 52.66%, respectively. However, Mn treatment significantly increased the ^−^OH level of sunflower roots (24.20%) but decreased it in leaf tissues (13.63%). Interestingly, CA treatment significantly decreased H_2_O_2,_ O_2_^−^, ^−^OH and MDA contents in the Mn-treated shoots and roots, except H_2_O_2_ content in Mn-treated leaves.

Application of ALA/SA could decrease most ROS in both leaf and root tissues of Mn- or Mn + CA-treated plants. For instance, ALA2 significantly reduced the O_2_^−^ content of Mn-treated leaves and roots by 47.44% and 26.54%, respectively. Interestingly, under Mn + CA treatment, foliar spray of the PGRs ALA or SA could significantly decrease the H_2_O_2_ content in roots, but significantly enhance the content of H_2_O_2_ in leaves (except ALA2 without a significant decline). ALA1 and ALA2 could dramatically decrease all of the ROS and MDA contents in Mn- or Mn + CA-treated sunflower roots. SA2 had significantly negative effects on the contents of O_2_^−^ and MDA in Mn + CA-treated leaves; however, the effects on roots were positive. Therefore, there were complex synergistic effects of citric acid combined with ALA/SA in response to manganese treatment in *Helianthus annuus*.

### 3.3. Antioxidant and Non-Antioxidant Enzyme Activities

Mn significantly increased most of the studied antioxidant enzymes’ activities, e.g., POD, CAT and APX in leaves and CAT and APX in roots (Figure 2). CA application to Mn-treated plants further activated the antioxidant enzymes mentioned above and SOD of Mn-treated leaves; however, the SOD activity in Mn-treated roots significantly declined after CA treatment. Under Mn treatment, all of the PGRs used in this experiment could significantly enhance most antioxidant enzymes’ activities in leaves, except APX and Mn + ALA1 treatment in POD, and the active effect of ALA2 was most obvious. Only SA2 significantly increased the SOD, POD and CAT activities in both leaf and root tissues of Mn-treated plants. Under Mn + CA stress, ALA1 and ALA2 significantly enhanced the SOD, POD and CAT activities in leaves and the SOD activity in roots. Interestingly, a high concentration of SA2 significantly increased the SOD and CAT activities in Mn + CA-treated roots, but significantly decreased those two enzymes’ activities in leaves. Moreover, the SOD and POD activities in leaves decreased obviously with SA concentration increasing from SA1 to SA2. However, PGRs could significantly decline the APX activity in both leaves and roots under Mn + CA treatment (Figure 2).

Mn treatment significantly increased the GSH content and GSH/GSSG ratio in both leaf and root tissues; however, it enhanced the GSH + GSSG content only in leaf tissues (Figure 3). ALA2 significantly increased the GSH + GSSG, GSH, and GSH/GSSG in Mn- or Mn + CA-treated leaves. However, the effect of SA1 was significant only on GSH + GSSG and GSH of Mn- or Mn + CA-treated leaves. In roots, CA could increase the GSH content of Mn-treated plants, but ALA or SA significantly decreased the GSH content of Mn + CA-treated roots. Interestingly, ALA1 or SA1 could significantly decrease the GSH + GSSG content of roots under Mn + CA treatment; however, the effect of SA2 was opposite. ALA2 or SA2 significantly decreased the GSH content and GSH/GSSG ratio in roots under Mn + CA stress (Figure 3).

### 3.4. Uptake and Translocation of Mn

The effects of ALA/SA combined with citric acid on Mn concentration, bioconcentration factor (BCF) and translocation factor (TF) in sunflowers under Mn treatment are presented in Table 3. The chelating agent CA treatment could significantly increase root Mn concentration by 94.25%; however, Mn transportation from roots to shoots was significantly inhibited, as revealed by the low Mn concentration in shoots after the application of CA. The trends in the bioconcentration factor (BCF) were similar to the Mn concentrations in roots and shoots after the application of CA on Mn-treated plants. Thus, the application of CA decreased the translocation factor (TF) of Mn in sunflower plants. ALA/SA could promote Mn absorption and significantly enhance the BCF of roots and shoots in Mn-treated sunflowers. Therefore, further application of ALA/SA could dramatically enhance the shoot Mn concentration and decrease the root Mn concentration in Mn + CA-treated plants. The highest shoot Mn concentration reached 819.34 mg kg^−1^ under the Mn + CA + ALA1 treatment, and the peak BCF of shoots was also observed in this combined treatment. However, the highest TF was found under the Mn + CA + SA2 treatment.

Considering the plant biomass, the effects of ALA/SA combined with citric acid on the Mn bioaccumulation quantity (BCQ) and removal efficiency (RE) in sunflowers under Mn treatment are revealed in Table 4. In Mn-treated plants, ALA/SA could significantly enhance shoot Mn BCQ. For instance, ALA2 or SA1 significantly increased shoot Mn BCQ (86.76%, 110.34%) and total BCQ (81.58%, 111.57%), respectively. Citric acid (CA) could significantly enhance root Mn BCQ; however, further application of ALA/SA dramatically decreased this effect. Interestingly, the trends in the shoot or total Mn BCQ were opposite to those in roots. For instance, CA application inhibited shoot Mn BCQ by 96.59% compared with Mn treatment alone. However, ALA/SA could recover this inhibition. For example, shoot Mn BCQ under Mn + CA + ALA1 was enhanced by 59.58 times compared with Mn + CA treatment alone. In contrast, root Mn BCQ was significantly decreased when Mn + CA-treated plants were sprayed with ALA/SA, although it was significantly higher than that of Mn treatment alone. In addition, ALA1, ALA2, SA1 and SA2 significantly increased Mn-removal efficiency (RE) by 36.84%, 84.21%, 115.79% and 73.68%, respectively, compared with Mn treatment alone. However, CA significantly decreased the RE of Mn-treated plants, while application of ALA/SA could significantly recover this depression.

### 3.5. Uptake of Other Elements

The concentrations of eleven elements in treated sunflower plants were detected to further evaluate the correlations among these elements, especially for the relationship between Mn absorption and other elements. The results of various elements concentration including metallic (Mn, Na, Mg, Al, P, K, Ca, Fe, Cu and Zn) and non-metallic (B) in sunflower shoots and roots are presented in Appendix A. The highest Mn concentration was observed in the root, and the lowest was in the shoot of Mn + CA-treated plants. The highest concentration was found for K in the Mn + CA + SA1-treated sunflower roots (66.57 g kg^−1^). Interestingly, from these results we found that K concentrations were higher than those of any other elements in both sunflower shoots and roots under all treatments. The concentrations of two other elements, Ca and P, were in the second and third position, respectively, following that of K in different sunflower tissues. In addition, the concentrations of Fe, Cu and Zn in shoots were significantly reduced under Mn + CA, which was consistent with the trends in Mn concentration (Appendix A). Therefore, the translocation of Fe, Cu and Zn elements from roots to shoots of sunflowers had close correlations with that of Mn.

### 3.6. Effects of ALA/SA on Cellular Organelles of Mn- and CA-Treated Plants

One of the main objectives of this study was to understand the cellular mechanism of Mn and CA’s effect in response to ALA/SA treatment. The ultrastructural changes in leaf mesophyll and root-tip cells under control and higher concentrations of ALA/SA combined with CA are illustrated in Figure 4 and Figure 5. The TEM micrographs of leaf mesophyll cells of the control contained a well-developed chloroplast with well-organized grana and thylakoid membrane system. The nucleus had a clear nuclear membrane with a nucleolus as well as mitochondria (Figure 4a). Under Mn treatment (0.8 g kg^−1^), chloroplasts were round with swollen thylakoids, dissolved membranes and a substantial increase in the number of plastoglobuli inside the stroma (Figure 4b). The application of a chelator (CA) along with Mn altered the cellular structure, which presented a clear cell wall, complete cell membrane as well a lens-shaped chloroplast that possessed well-organized grana and thylakoids with a few dense plastogobuli (Figure 4c). ALA- or SA-treated plants showed reduced grana and stroma thylakoids with ruptured membranes and increased plastoglobuli accumulation in the chloroplast (Figure 4d,g). Mesophyll cells of ALA/SA + Mn-treated plants improved the cell wall and cell membrane structures as compared with Mn-treated plants. ALA + Mn-treated plants contained a clear nucleolus in the nucleus (Figure 4e). Similarly, SA + Mn-treated plants possessed a fusiform chloroplast with well-organized cell wall and chloroplasts as well as more visible grana and mitochondria (Figure 4h). Further, spraying with ALA added significant cellular damage to Mn + CA, where the chloroplast was deformed with a loose thylakoid membrane system and two large sizes of starches as well as a considerable amount of plastoglobuli inside the stroma (Figure 4f). Compared with Mn + CA treatment alone, the lamellar structure in the SA cell was relatively loose and the chloroplast envelope was intact with a much larger size of starch grains (Figure 4i).

Different treatments also significantly affected ultrastructural changes in root-tip cells (Figure 5). Control plants had typical mature cells with a definite cell wall, containing a complete nucleus and clear mitochondria, and the nucleus had a nuclear membrane and nucleolus (Figure 5a). The Disappearance of the nucleolus, deepening of the nuclear matrix and deformation of the cell wall were some of the other obvious changes observed under Mn-treated cells (Figure 5b). There was a significant increase in the number and area of vacuoles in root-tip cells. Moreover, the nucleus was severely collapsed and the nucleolus became invisible in the Mn + CA treatment (Figure 5c). Compared with the control, no significant changes were noticeable under the treatment of AL. The cell contained a complete nucleus and numerous mitochondria with smooth cell walls and cristae (Figure 5d). However, numerous ultrastructural alterations were noticed in root meristematic cells of the plants exposed to Mn combined with 20 mg L^−1^ ALA. There was an increased Mn deposition in the vacuoles, and most of the organelles disappeared (Figure 5e). The application of ALA significantly improved the cell shape and showed a clear and intact nucleus in Mn + CA treated cells (Figure 5f). Further, applying SA alone resulted in several plastoglobuli inside the stroma and mitochondria and also caused a deformed nucleus with an unclear nucleolus (Figure 5g). SA + Mn-treated plants presented cells with a large vacuole and deformed nucleus as well as a dissolved cell wall (Figure 5h). Similarly, CA reversed the negative effects of Mn + SA on nuclear structure, including the nuclear membrane and nucleolus (Figure 5i).

### 3.7. Effects of ALA/SA on Gene Expression under Mn and CA Treatments

We then explored whether 20 key regulatory genes, including antioxidant enzymes, metal transporters, metal tolerance protein, resistance mechanism-related proteins and genes in the key metabolic pathways, contribute to the changes in roots and leaves under different treatments with RT-qPCR (Figure 6, Appendix A). The data are presented as a heatmap in clusters using log_2_ fold-change values (Figure 6a,b). In sunflower leaves, the metal tolerance protein B1 (*MTPB1*) and *lox* genes were significantly up-regulated (log_2_ based fold change >2) by Mn alone or with CA (Figure 6a). Approximately half of the studied genes, including *MTPB1*, *def*, iron regulated transporter 1 (*IRT1*), *HMA2*, *lox*, *CAX2*, *Mn-SOD*, *ZIP6*, *VIP1* and *NPR1*, were significantly up-regulated (log_2_ based fold change >2) in leaves under the Mn + SA2 treatment. Moreover, the expression levels of the *IRT1*, *MTPB1*, *lox*, *def*, *HMA2*, natural resistance-associated macrophage protein 3 (*NRAMP3*) and *ZIP6* genes were significantly up-regulated (log_2_ based fold change >2) by Mn + CA. ALA2 further promoted the expression of the *def* gene under the Mn + CA treatment (Figure 6a). In sunflower roots, the auxin-induced protein PCNT115 (*HaAC1*) gene was significantly up-regulated (log_2_ based fold change >2) by Mn alone or with CA + SA and *MTPB1* was significantly up-regulated (log2 based fold change >2) by Mn alone, Mn +SA and CA + SA treatment. However, most of the studied genes (e.g., *POX*, *NRAMP3*, *ECA1*, *MT2* and *XTH9*) were significantly down-regulated (log_2_ based fold change <1) under different treatments compared with the control in sunflower roots (Figure 6b).

Based on RT-qPCR profiling, different relative expression patterns were identified for different treatments on sunflower growth (Appendix A). During Mn treatment, significantly up-regulated changes were detected for the relative expression of the *MTPB1*, *def*, *lox* and *HMA2* genes in leaves compared with the control, which was normalized as “1”. For example, the JA-responsive gene *lox* was 9.05-fold higher in Mn-treated leaves than in the control. However, most of the genes in leaves, e.g., *POX*, *HaAC1*, *NPR1*, *OPT6*, *CAX2*, *ZIP6*, *VIP1*, *MTP11*, *NRAMP2*, *NRAMP3*, *MT2*, *ECA3* and *XTH,* were significantly down-regulated in response to Mn treatment. In contrast, these genes were significantly up-regulated in response to ALA2 under Mn treatment. The application of CA significantly up-regulated the expression levels of the *IRT1*, *CAX2*, *ZIP6*, *VIP1*, *MTPB1*, *MTP11*, *NRAMP3*, *def*, *lox*, *HMA2*, *ECA3* and *XTH* genes compared with Mn alone. SA promoted expression levels of most of the studied genes (including *ECA1* and *XTH*) except for *NRAMP2* and *MT2* in the Mn + CA treatment (Appendix A). In sunflower roots, the *NPR1* and *MTPB1* genes were significantly up-regulated while *POX*, *NRAMP3*, *MT2*, *ECA1* and *MTPB1* were significantly down-regulated under all treatments (Mn, CA, ALA and SA) compared with their respective controls. Other genes, such as *Mn-SOD*, *OPT6*, *CAX2*, *VIP1*, *NRAMP2* and *HMA2*, showed no significant difference in response to any treatments used in this study in sunflower roots (Appendix A).

### 3.8. Correlations among Different Parameters

Pearson correlation analysis was performed to determine the correlations among the plant growth, oxidative damage, element uptake, metal translocation, bioavailable metal content and sunflower removal efficiency (Appendix A). The shoot biomass was negatively correlated with H_2_O_2_ in leaves (*p* < 0.01) and shoot Mn concentration (*p* < 0.01) but it was positively correlated with APX in leaves (*p* < 0.05). Although the root biomass had a positive correlation (*p* < 0.01) with root Mn concentration, it was negatively correlated with Mn translocation factor (*p* < 0.01). Furthermore, the root Mn concentration was significantly (*p* < 0.01) negatively correlated with Mn translocation factor but had no significant correlations with removal efficiency. Moreover, the shoot Mn concentration was significantly (*p* < 0.01) positively correlated with Mn translocation factor and removal efficiency; however, it was negatively correlated (*p* < 0.05) with root Mn concentration (Appendix A). Further, the shoot Mn concentration was positively correlated with the concentrations of B, Al, P, Ca, Fe, Cu, Zn (*p* < 0.01) and Mg (*p* < 0.05) (Appendix A). The absorption of Cu had a significant concentration with Zn (*p* < 0.01). There were significant correlations between Ca and three other elements (Fe, Cu and Zn). However, shoot K concentration only had a significant correlation with that of Mg (*p* < 0.05). In roots, the Mn concentration had a positive correlation with Mg and Cu (*p* < 0.01) only. The root B concentration was positively correlated with that of Ca (*p* < 0.01) and P (*p* < 0.05). In addition, root Al absorption had a significant correlation with Fe (*p* < 0.01) (Appendix A).

## 4. Discussion

Manganese (Mn) is an essential element for plant growth, but its availability differs greatly in space and time, depending largely on the nature and amount of the Mn minerals present and on the soil’s pH and redox potential [5]. A low Mn concentration promotes plant growth as an important micronutrient, while a high Mn concentration in soils can induce toxicity to plants due to the excessive application of acidic fertilizer [5]. The knowledge of Mn’s toxic effects on the plant may contribute to a better understanding of the toxicity mechanism and plant responses [7,37]. In the present study, there was a significant increase in sunflower growth attributes, i.e., shoot biomass (fresh and dry weight) and plant height, under 0.8 g kg^−1^ Mn. CA, one kind of low-molecular-weight organic acid (LMWOA), could activate metals in the soil and accelerate their uptake by plants. The highest biomass was observed under Mn + CA stress. Moreover, the application of CA increased the absorption of Mn, indicating that CA could promote sunflower growth under 0.8 g kg^−1^ Mn, but still does not reach the threshold of Mn injury to sunflower plants. The threshold of Mn injury is highly dependent on plant species or genotypes [3]. PGRs, i.e., 10 mg L^−1^ ALA or 50 mg L^−1^ SA, could also accelerate the growth of sunflowers under 0.8 g kg^−1^ Mn treatment. However, the synergistic effects of CA combined with ALA/SA on manganese phytoextraction in *Helianthus annuus* L. were complex. The application of ALA/SA decreased the biomass of sunflowers under Mn + CA treatment, implying that PGRs could further regulate sunflower growth by adjusting the Mn concentration in the plants. Similar regulation effects of ALA have been reported by Ali et al. [10,11] in *Brassica napus* and Xu et al. [9] in sunflowers under Cd stress.

For Mn uptake and translocation, CA could increase the bioavailability of Mn in the soil, as revealed by the increased Mn concentration in the roots of sunflowers under Mn + CA treatment compared to Mn treatment alone. This result was consistent with Najeeb et al. [7], who reported the effects of CA in the Mn-treated wetland plant *Juncus effusus*. However, the application of CA in the soil significantly inhibited the transport of Mn from roots to shoots in sunflowers. This research revealed that the activity of antioxidant enzymes, including SOD, POD and CAT, in sunflower leaves was significantly increased by CA under Mn treatment to scavenge ROS. Xiong et al. [38] reported that exogenous H_2_O_2_ treatment in the roots of rice not only inhibited root elongation, but also increased the root diameter. In this study we found that the H_2_O_2_ levels were significantly increased in both sunflower leaves and roots by Mn alone, but CA significantly reduced the level of H_2_O_2_ in sunflower leaves while further significantly increasing it in roots under Mn treatment. The RT-qPCR analysis also exhibited that the *IRT1*, *CAX2*, *def* and *ECA3* genes were up-regulated by Mn. CA application significantly down-regulated these genes’ expression levels in sunflower roots, but showed the opposite trend in sunflower leaves. It might be that the depressed Mn transport protein inhibited Mn^2+^ transportation from roots to leaves under the effects of CA. This study also further verified the effects of ALA/SA combined with citric acid on manganese absorption in sunflowers with TEM micrographs of leaf mesophyll and root cells. Electron microscopy helps to assess damage at the tissue and ultrastructural levels. The mesophyll cells were significantly damaged under Mn + CA + ALA compared with Mn + CA treatment, while the ultrastructure of root cells showed that ALA improved the damage by Mn with CA. This result also illustrated that CA significantly improved the accumulation of Mn in roots, which is opposite to Najeeb et al. [20], who found that CA improved the root-cell shape under Cd stress in *J. effusus* plants. However, ALA/SA promoted the transportation of Mn from roots to shoots and contributed to the maximum cell-structure damage in leaf mesophyll in sunflowers.

ALA and SA significantly promoted the Mn absorption of shoots under Mn alone or with CA. ALA/SA significantly increased the root Mn concentration under Mn treatment. However, the application of ALA/SA significantly reduced the absorption of Mn in sunflower roots under Mn + CA stress. It might be that ALA/SA could act as a pump transporting Mn ions from roots to shoots. The Mn bioaccumulation quantity (BCQ) and removal efficiency (RE) in sunflowers further demonstrated the effects of ALA/SA under Mn + CA treatment. These results were similar to the findings of Cd transport mediated by ALA in sunflowers [9]. Previous studies found that foliage spray of ALA could protect cell membranes from lipid peroxidation [39] by triggering the activities of antioxidant enzymes and scavenging ROS [10,11,12]. Moreover, a suitable concentration of ALA treatment could be a benefit for chlorophyll biosynthesis and the consequent accumulation of carbohydrates [12]. Thus, suitable PGRs have positive effects on sunflower growth by regulating the uptake of Mn. However, with the increase in Mn concentration in the shoot from the Mn + CA to the Mn + CA + ALA/SA treatment, the biomass declined significantly. This might be due to excessive Mn content in the sunflower shoot tissues. Thus, we draw a schematic diagram (Figure 7) to illustrate the dual (beneficial/toxic) effects of ALA/SA combined with CA on the phenotypic, physio-biochemical, ultrastructure and molecular processes in Mn-treated *H. annuus* plants.

Candidate key gene identification is important for further mechanism studies [40]. In the present investigation, different treatments (i.e., ALA, SA, Mn and CA) induced significant changes in the expression of the *Mn-SOD* gene. This gene regulates adaptive mechanisms and tolerance to environmental stress [3]. Further, the *NRAMP2* and *NRAMP3* genes were down-regulated under Mn treatment in both leaves and roots compared with their respective controls. This suggests that an optimal Mn concentration in root tissues may be maintained via an exclusion mechanism through down-regulation of this gene [41]. Ionic uptake and transport inside the plant body are facilitated by specific transporters [42]. Metal ions are essential cofactors for biological processes, including oxidative phosphorylation, gene regulation and free-radical homeostasis [43]. A previous study summarized that *AtIRT1*, *CAX2* and *Nramp* were identified as Mn^2+^ uptake transporters [3]. In addition, Ca^2+^-dependent successive phosphorylation of the vacuolar transporter MTP8 by CBL2/3-CIPK3/9/26 and CPK5 is critical for Mn homeostasis in *Arabidopsis* [44]. Natural resistance-associated macrophage proteins (NRAMPs) regulate the transport of various ions including Mn under low-Mn conditions in plants [45,46]. However, it is unclear whether this is restricted to a difference at the transcriptional level (plant, tissue or cell-type specific) or whether NRAMP proteins are functionally divergent between hyperaccumulating and non-hyperaccumulating plants [47]. In this study, *IRT1* gene expression in Mn-treated leaves was induced by CA application and reduced by ALA/SA. IRT1 plays an important role in Mn uptake [48] and variable induction of *IRTI* in response to chelators and PGRs suggested their differential regulatory mechanism for adapting to high-Mn environments. In addition, *NPR1* is a key regulator of the SA signal pathway and we found that the expression level of *NPR1* was significantly increased after the application of SA under Mn stress as compared with Mn + CA treatment in sunflower roots.

Further, Pearson correlation analysis has been widely used to evaluate the relationship between plant phenotype and measured parameters [49,50]. In this study, the shoot Mn concentration presented significant (*p* < 0.01) negative correlations with the shoot biomass but was significantly (*p* < 0.05) positively correlated with the H_2_O_2_ content in leaves. These results are similar to those of Najeeb et al. [7] and support the hypothesis proposed by Lian et al. [51], who found that decreased shoot biomass and increased root oxidative damage could be induced by severe phytotoxicity of high-Cd stress. In this study, we use the strategy of ALA/SA combined with CA to regulate Mn absorption. The results showed that the root biomass was enhanced with an increase in the Mn concentration in roots (Appendix A). That means that the root Mn concentration displayed a significant (*p* < 0.01) positive correlation with the root biomass. The results were due to Mn’s status as an essential micronutrient, different from the heavy metal Cd, which can have a severe impact on plant growth [52]. Previous studies revealed that phosphate deprivation will decrease metal uptake by competitively increasing other metal uptake and accumulation. Excess Mn in the growth medium can interfere with the absorption, translocation and utilization of other mineral elements such as Ca, Mg, Fe and P [37]. Moreover, Fe is essential for plant growth and development, and an accumulation of Fe could enhance rice’s tolerance to Cd stress [53]. This study of the correlation between Mn and other elemental contents proved that Mn concentration was significantly correlated with most elements in the shoots but only significantly correlated (*p* < 0.01) with Mg and Cu in the roots.

## 5. Conclusions

This study was conducted to evaluate the promotive role of ALA/SA combined with citric acid on Mn absorption in *Helianthus annuus* (Figure 7). The results indicated that the chelating agent CA enhanced the effects of Mn stress on the sunflower by increasing the plant biomass but inhibiting photosynthesis. Moreover, CA significantly increased MDA, O^2−^ and H_2_O_2_ contents, elevating the oxidative damage to sunflowers. The application of ALA/SA significantly enhanced the SOD, POD and CAT activities and GSH content in sunflower plants. Moreover, CA activated Mn in the soil and significantly increased the absorption of Mn in sunflower roots; however, it did not accelerate the translocation efficiency. It suggests that perhaps Mn was isolated in root tissue. Furthermore, Mn, CA and ALA/SA, to some extent, significantly induced ultrastructural changes and Mn-regulated gene expression patterns under various treatments. Like a pump, ALA/SA regulated the translocation efficiency and promoted transportation, guiding the further study of mechanisms of Mn regulation in plants. This strategy of synergistic application with ALA/SA and CA sheds light on the mechanism of plant growth by regulating manganese absorption, which provides a reference for the mechanism of Mn absorption in plants in the future.

## Figures and Tables

**Figure 1 antioxidants-12-00580-f001:**
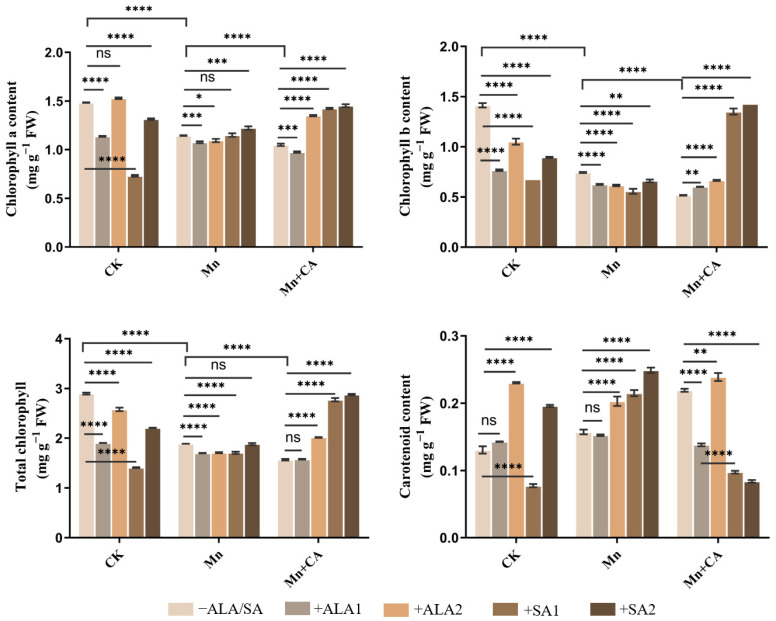
Effects of 5-aminolevulinic acid or salicylic acid (ALA/SA) combined with citric acid (CA) on chlorophyll a, chlorophyll b, total chlorophyll and carotenoid contents under manganese (Mn) treatment in sunflower plants. CA means 20 mmol kg^−1^ citric acid treatment. ALA1/ALA2 indicates spraying with 10/20 mg L^−1^ ALA. SA1/SA2 indicates spraying with 50/100 mg L^−1^ SA. CK, control plants without any treatment; Mn means that plants were exposed to 0.8 g kg^−1^ Mn for 7 days; Mn + CA indicates plants treated with CA for 7 days after Mn treatment. Mn + CA + ALA/SA treatment indicates plants treated with ALA/SA for one week after CA treatment. Note: Data are the means of three replicates (mean ± SE). * indicates a significant difference at *p*-value  <  0.05, ** *p*-value  <  0.01, *** *p*-value  <  0.001, **** *p*-value  <  0.0001; ns indicates no significant difference according to Tukey’s multiple comparisons test.

**Figure 2 antioxidants-12-00580-f002:**
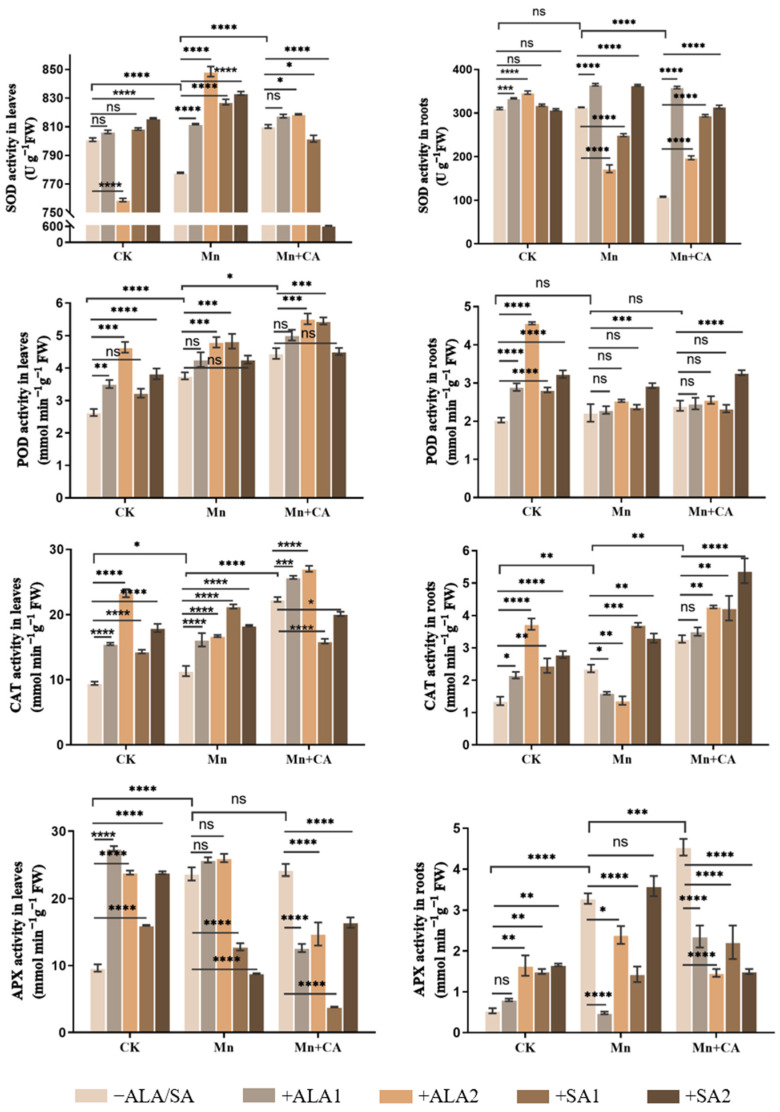
Effects of 5-aminolevulinic acid or salicylic acid (ALA/SA) combined with citric acid (CA) on antioxidant enzyme activities in the leaves and roots of manganese (Mn)-treated sunflower plants. CA means 20 mmol kg^−1^ citric acid treatment. ALA1/ALA2 indicates spraying with 10/20 mg L^−1^ ALA. SA1/SA2 indicates spraying with 50/100 mg L^−1^ SA. CK, control plants without any treatment; Mn means that plants were exposed to 0.8 g kg^−1^ Mn for 7 days; Mn + CA indicates plants treated with CA for 7 days after Mn treatment. Mn + CA + ALA/SA treatment indicates plants treated with ALA/SA for one week after CA treatment. Note: Data are the means of three replicates (mean ± SE). * indicates a significant difference at *p*-value  <  0.05, ** *p*-value  <  0.01, *** *p*-value  <  0.001, **** *p*-value  <  0.0001; ns indicates no significant difference according to Tukey’s multiple comparisons test.

**Figure 3 antioxidants-12-00580-f003:**
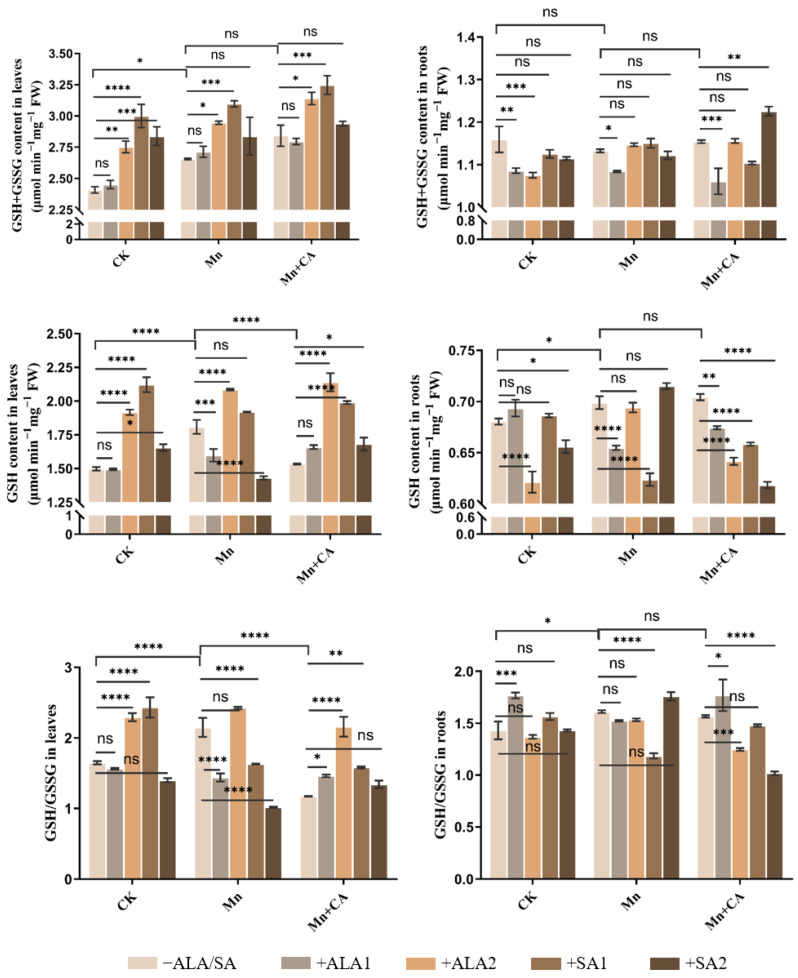
Effects of 5-aminolevulinic acid or salicylic acid (ALA/SA) combined with citric acid (CA) on total glutathione content (GSH + GSSG), glutathione reduced (GSH) and glutathione reduced/glutathione oxidized (GSH/GSSG) ratio in the leaves and roots of manganese (Mn)-treated sunflower plants. CA means 20 mmol kg^−1^ citric acid treatment. ALA1/ALA2 indicates spraying with 10/20 mg L^−1^ ALA. SA1/SA2 indicates spraying with 50/100 mg L^−1^ SA. CK, control plants without any treatment; Mn means that plants were exposed to 0.8 g kg^−1^ Mn for 7 days; Mn + CA indicates plants treated with CA for 7 days after Mn treatment. Mn + CA + ALA/SA treatment indicates plants treated with ALA/SA for one week after CA treatment. Note: Data are the means of three replicates (mean ± SE). * indicates a significant difference at *p*-value  <  0.05, ** *p*-value  <  0.01, *** *p*-value  <  0.001, **** *p*-value  <  0.0001; ns indicates no significant difference according to Tukey’s multiple comparisons test.

**Figure 4 antioxidants-12-00580-f004:**
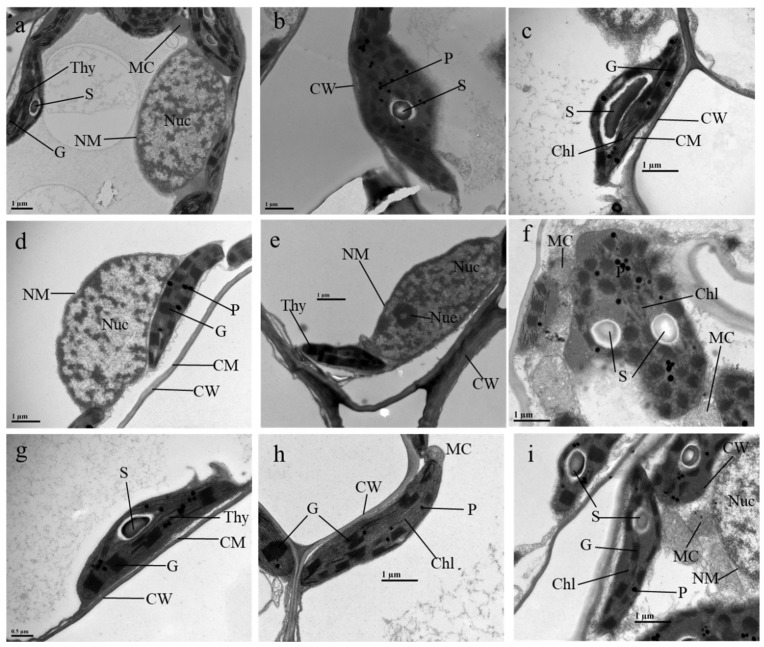
Effects of 5-aminolevulinic acid or salicylic acid (ALA/SA) combined with citric acid (CA) on electron micrographs of leaf mesophyll cells of manganese (Mn)-treated sunflower plants. (**a**–**c**) represent electron micrographs of mesophyll cells of sunflowers under CK, Mn and Mn + CA treatment, respectively; (**d**–**f**) represent electron micrographs of mesophyll cells of sunflowers under ALA, Mn + ALA and Mn + CA + ALA treatment, respectively; (**g**–**i**) represent electron micrographs of mesophyll cells of sunflowers under SA, Mn + SA and Mn + CA + SA treatment, respectively. CK, control plants without any treatment; Mn, plants exposed to 0.8 g kg^−1^ of manganese for 7 days; CA means 20 mmol kg^−1^ citric acid treatment. ALA/SA indicates spraying with 20/100 mg L^−1^ ALA/SA. Mn + CA, plants treated with CA for 7 days after Mn treatment. Mn + CA + ALA/SA treatment indicates plants treated with ALA/SA for one week after CA treatment. Plates explanation: MC, mitochondria; Chl, chloroplasts; NM, nuclear membrane; Thy, thylakoid; Nuc, nucleus; Nue, nucleolus; CW, cell wall; CM, cell membrane; P, plastoglobuli; S, starch; G, grana; Vac, vacuole. (**a**,**b**,**d**,**e**)—×20,000, (**c**,**f**,**h**,**i**)—×25,000, (**g**)—×30,000.

**Figure 5 antioxidants-12-00580-f005:**
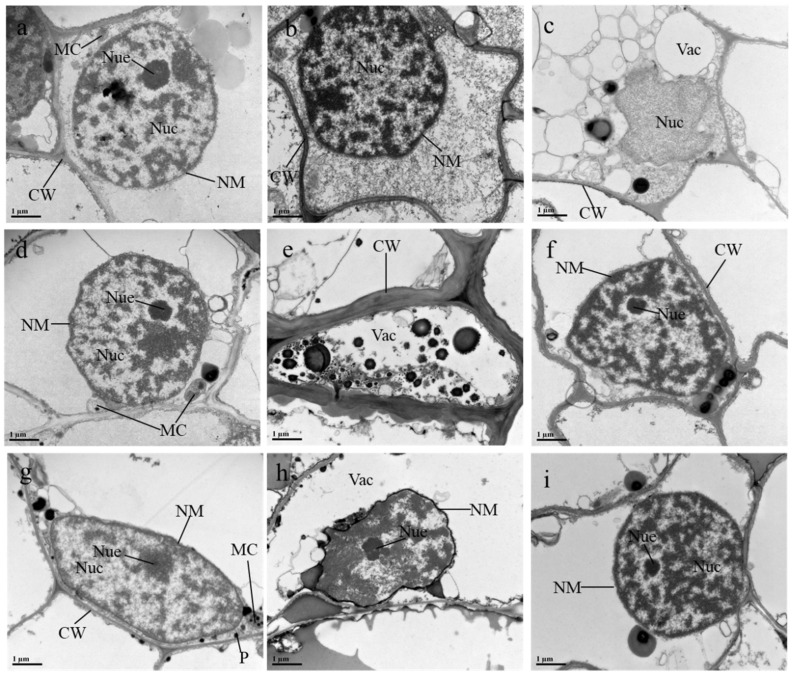
Effects of 5-aminolevulinic acid or salicylic acid (ALA/SA) combined with citric acid (CA) on electron micrographs of root cells of manganese (Mn)-treated sunflower plants. (**a**–**c**) represent electron micrographs of root cells of sunflowers under CK, Mn and Mn + CA treatment, respectively; (**d**–**f**) represent electron micrographs of root cells of sunflowers under ALA, Mn + ALA and Mn + CA + ALA treatment, respectively; (**g**–**i**) represent electron micrographs of root cells of sunflowers under SA, Mn + SA and Mn + CA + SA treatment, respectively. CK, control plants without any treatment; Mn, plants exposed to 0.8 g kg^−1^ Mn for 7 days; CA means 20 mmol kg^−1^ citric acid treatment. ALA/SA indicates spraying with 20/100 mg L^−1^ ALA/SA. Mn + CA, plants treated with CA for 7 days after Mn treatment. Mn + CA + ALA/SA treatment indicates plants treated with ALA/SA for one week after CA treatment. Plates explanation: MC, mitochondria; NM, nuclear membrane; Nuc, nucleus; Nue, nucleolus; CW, cell wall; CM, cell membrane; P, plastoglobuli. (**a**~**i**)—×20,000.

**Figure 6 antioxidants-12-00580-f006:**
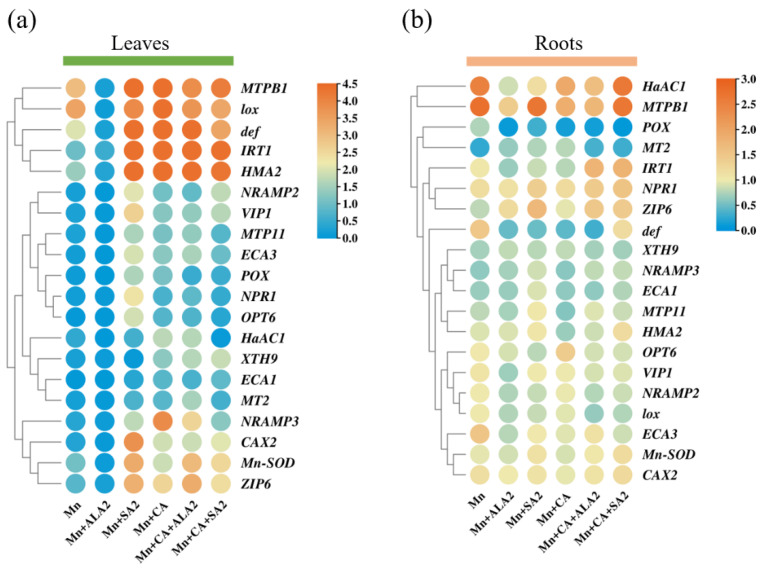
Heatmap representation of the effects of 5-aminolevulinic acid or salicylic acid (ALA/SA) combined with citric acid (CA) on tolerance-related gene expression patterns in the leaves (**a**) and roots (**b**) of manganese (Mn)-treated sunflower plants. Mn, plants exposed to 0.8 g kg^−1^ Mn for 7 days; Mn + CA, plants treated with CA for 7 days after Mn treatment; Mn + ALA2/SA2, plants treated with 20/100 mg L^−1^ ALA/SA for one week after Mn treatment; Mn + CA + ALA2/SA2, plants treated with 20/100 mg L^−1^ ALA/SA for one week after CA treatment. The relative expression levels are calculated with the 2^−ΔΔCt^ method compared with that of *EF-1α* with three replicates. Every control of the expression level was normalized as “1” and log_2_ fold change between two treatments is used to present expression changes.

**Figure 7 antioxidants-12-00580-f007:**
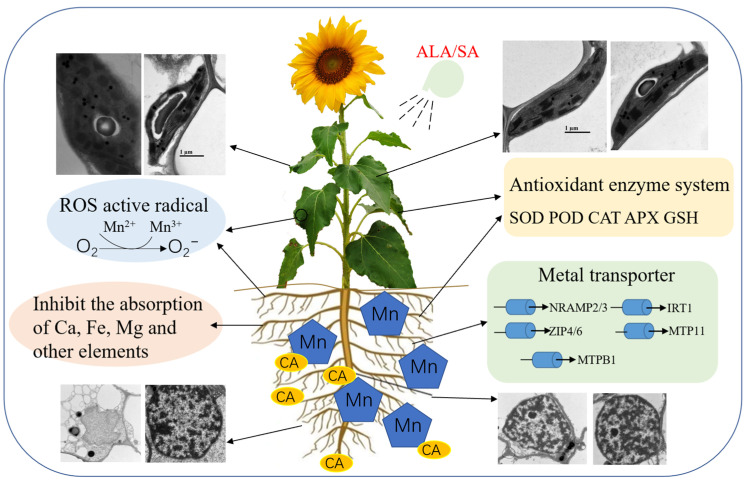
A model figure to illustrate the dual (beneficial/toxic) effects of 5-aminolevulinic acid or salicylic acid (ALA/SA) combined with citric acid (CA) on manganese (Mn)-treated *Helianthus annuus* L. seedlings.

**Table 1 antioxidants-12-00580-t001:** Effects of 5-aminolevulinic acid or salicylic acid (ALA/SA) combined with citric acid (CA) on fresh weight (FW) and dry weight (DW) of shoot, root and plant height of sunflower plants under manganese (Mn) treatment.

Treatments	Shoot (g)		Root (g)		Plant Height (cm)
	FW	DW	FW	DW	
CK	3.58 ± 0.06 gh	0.5198 ± 0.04 fg	1.602 ± 0.02 e	0.1169 ± 0.004 cd	36.67 ± 0.16 ef
ALA1	4.22 ± 0.03 cd	0.6223 ± 0.01 bcd	1.737 ± 0.01 de	0.1358 ± 0.017 cd	38.66 ± 0.88 d
ALA2	3.30 ± 0.07 h	0.4636 ± 0.01 h	1.650 ± 0.05 e	0.1236 ± 0.002 cd	33.38 ± 0.47 h
SA1	3.98 ± 0.14 def	0.5705 ± 0.02 def	1.741 ± 0.07 de	0.1365 ± 0.005 cd	36.17 ± 0.44 f
SA2	4.19 ± 0.03 cd	0.6063 ± 0.02 bcd	1.763 ± 0.03 cde	0.1380 ± 0.007 cd	39.17 ± 0.17 cd
Mn	4.07 ± 0.05 de	0.5740 ± 0.01 de	1.627 ± 0.01 e	0.1195 ± 0.004 cd	38.15 ± 0.67 d
Mn + ALA1	5.14 ± 0.08 ab	0.6544 ± 0.01 b	2.043 ± 0.17 bcd	0.1405 ± 0.007 bcd	40.83 ± 0.17 ab
Mn + ALA2	5.26 ± 0.01 ab	0.6584 ± 0.01 b	1.544 ± 0.04 e	0.1094 ± 0.003 de	42.00 ± 0.29 a
Mn + SA1	4.98 ± 0.03 b	0.6314 ± 0.04 bc	2.280 ± 0.13 ab	0.1689 ± 0.024 ab	40.21 ± 0.27 bc
Mn + SA2	3.71 ± 0.05 fg	0.5258 ± 0.01 efg	1.641 ± 0.02 e	0.1223 ± 0.008 cd	35.83 ± 0.33 fg
Mn + CA	5.40 ± 0.25 a	0.7779 ± 0.01 a	2.480 ± 0.18 a	0.1746 ± 0.001 a	40.33 ± 0.17 bc
Mn + CA + ALA1	3.50 ± 0.08 gh	0.5146 ± 0.01 g	1.640 ± 0.12 e	0.1220 ± 0.002 cd	35.33 ± 0.33 fg
Mn + CA + ALA2	4.14 ± 0.06 cd	0.5868 ± 0.01 cd	2.117 ± 0.10 bc	0.1437 ± 0.008 bc	36.67 ± 0.44 ef
Mn + CA + SA1	4.47 ± 0.24 c	0.6292 ± 0.01 bc	2.103 ± 0.19 bcd	0.1425 ± 0.003 bc	37.83 ± 0.73 de
Mn + CA + SA2	3.78 ± 0.07 efg	0.5313 ± 0.02 efg	1.150 ± 0.24 f	0.0857 ± 0.013 e	34.50 ± 0.50 gh

Note: Each value is the mean of three individual replicates. Means followed by the same letter are not significantly different at *p* < 0.05. CK is the control plants without any treatment. CA means 20 mmol kg^−1^ citric acid treatment. ALA1/ALA2 indicates spraying with 10/20 mg L^−1^ 5-aminolevulinic acid. SA1/SA2 indicates spraying with 50/100 mg L^−1^ salicylic acid. Mn means that plants exposed to 0.8 g kg^−1^ of manganese for 7 days; Mn + CA indicates plants treated with CA for 7 days after Mn treatment. Mn + CA + ALA/SA treatment indicates plants treated with ALA/SA for one week after Mn + CA treatment.

**Table 2 antioxidants-12-00580-t002:** Effects of 5-aminolevulinic acid or salicylic acid (ALA/SA) combined with citric acid on hydrogen peroxide (H_2_O_2_), superoxide radical (O_2_^−^), hydroxyl ion (^−^OH) and malondialdehyde (MDA) contents in sunflower leaves and roots under manganese (Mn) treatment.

Treatments	H_2_O_2_ (μmol g^−1^ FW)		O_2_^−^ (nmol min^−1^ g^−1^ FW)		^−^OH (μmol g^−1^ FW)		MDA (nmol g^−1^ FW)	
	Leaf	Root	Leaf	Root	Leaf	Root	Leaf	Root
CK	17.54 ± 0.21 hi	14.75 ± 0.68 f	6.33 ± 0.16 h	3.49 ± 0.02 d	98.25 ± 0.28 b	32.35 ± 0.21 d	39.43 ± 0.65 e	14.09 ± 0.28 fg
ALA1	20.66 ± 1.78 fg	19.61 ± 0.34 bc	9.22 ± 1.26 cde	4.11 ± 0.02 a	62.20 ± 0.61 k	34.23 ± 0.28 c	37.90 ± 0.47 f	12.90 ± 0.32 fgh
ALA2	19.35 ± 0.71 gh	12.65 ± 0.09 g	6.82 ± 0.36 gh	3.25 ± 0.03 e	82.39 ± 0.55 ef	26.52 ± 0.09 h	47.04 ± 0.16 a	19.46 ± 1.55 bc
SA1	28.12 ± 1.63 a	13.36 ± 0.08 g	7.72 ± 0.34 fg	3.62 ± 0.03 cd	83.61 ± 0.75 de	34.29 ± 0.11 c	42.88 ± 0.54 c	17.10 ± 0.19 de
SA2	16.56 ± 0.17 i	18.53 ± 0.59 d	10.38 ± 0.57 bc	3.14 ± 0.01 ef	77.79 ± 0.56 g	30.43 ± 0.47 ef	37.50 ± 0.23 f	16.45 ± 0.49 e
Mn	24.84 ± 0.23 cd	17.27 ± 0.25 e	14.44 ± 0.05 a	4.07 ± 0.04 a	84.86 ± 0.56 d	40.18 ± 0.56 a	44.62 ± 0.31 b	21.51 ± 0.17 a
Mn + ALA1	20.99 ± 0.86 fg	15.29 ± 0.18 f	9.72 ± 0.10 cd	3.03 ± 0.01 fg	87.91 ± 0.33 c	34.18 ± 0.54 c	37.37 ± 0.78 f	17.42 ± 1.22 de
Mn + ALA2	21.86 ± 0.19 ef	15.28 ± 0.03 f	7.59 ± 0.25 fgh	2.99 ± 0.07 fg	72.05 ± 0.37 i	28.71 ± 0.60 g	45.70 ± 0.64 b	19.35 ± 0.19 bc
Mn + SA1	21.66 ± 0.45 ef	18.36 ± 0.64 d	11.39 ± 0.14 b	3.89 ± 0.02 b	106.21 ± 0.06 a	36.43 ± 0.28 b	41.40 ± 0.31 d	12.47 ± 0.57 fgh
Mn + SA2	22.13 ± 0.26 ef	21.05 ± 0.41 a	7.51 ± 0.03 fgh	3.60 ± 0.11 cd	71.89 ± 0.43 i	35.20 ± 0.28 c	33.78 ± 0.32 g	12.26 ± 1.45 gh
Mn + CA	19.54 ± 0.65 gh	19.98 ± 0.14 b	8.39 ± 0.08 ef	3.90 ± 0.01 b	76.77 ± 0.20 g	34.18 ± 0.28 c	41.40 ± 0.22 d	20.86 ± 0.22 ab
Mn + CA + ALA1	25.93 ± 0.17 bc	18.64 ± 0.16 cd	7.94 ± 0.03 fg	3.57 ± 0.02 cd	72.16 ± 0.16 i	28.61 ± 0.19 g	37.10 ± 0.31 f	18.49 ± 0.11 cd
Mn + CA + ALA2	17.89 ± 0.47 hi	16.75 ± 0.31 e	6.91 ± 0.09 gh	2.96 ± 0.07 g	66.11 ± 0.28 j	31.02 ± 0.28 e	32.68 ± 0.38 g	16.68 ± 0.78 de
Mn + CA + SA1	23.28 ± 0.60 de	17.36 ± 0.02 e	7.41 ± 0.08 fgh	3.12 ± 0.04 ef	81.00 ± 0.93 f	40.39 ± 0.21 a	40.32 ± 0.54 de	14.25 ± 0.38 f
Mn + CA + SA2	27.36 ± 0.53 ab	15.69 ± 0.07 f	8.54 ± 0.21 def	3.68 ± 0.05 c	74.79 ± 0.57 h	29.89 ± 0.19 f	45.16 ± 0.543 b	12.10 ± 0.48 h

Note: Each value is the mean of three individual replicates. Means followed by the same letter are not significantly different at *p* < 0.05. CK is the control plants without any treatment. CA means 20 mmol kg^−1^citric acid treatment. ALA1/ALA2 indicates spraying with 10/20 mg L^−1^ 5-aminolevulinic acid. SA1/SA2 indicates spraying with 50/100 mg L^−1^ salicylic acid. Mn means that plants exposed to 0.8 g kg^−1^ of manganese for 7 days; Mn + CA indicates plants treated with CA for 7 days after Mn treatment. Mn + CA + ALA/SA treatment indicates plants treated with ALA/SA for one week after Mn + CA treatment.

**Table 3 antioxidants-12-00580-t003:** Effects of 5-aminolevulinic acid or salicylic acid (ALA/SA) combined with citric acid (CA) on manganese (Mn) concentration, bioconcentration factor (BCF), translocation factor (TF) in sunflower under Mn treatment.

Treatments	Shoot Mn Concentration (mg kg^−1^)	Root Mn Concentration (mg kg^−1^)	BCF of Shoots	BCF of Roots	TF
Mn	355.15 ± 29.10 d	246.04 ± 21.92 ef	0.44 ± 0.036 d	0.31 ± 0.03 ef	1.44 ± 0.12 f
Mn + ALA1	397.88 ± 29.05 d	391.23 ± 34.17 b	0.50 ± 0.036 d	0.49 ± 0.04 b	1.02 ± 0.07 g
Mn + ALA2	578.25 ± 34.98 c	391.37 ± 5.94 b	0.72 ± 0.04 c	0.49 ± 0.01 b	1.48 ± 0.09 f
Mn + SA1	679.07 ± 35.97 bc	383.20 ± 17.88 b	0.85 ± 0.04 bc	0.48 ± 0.02 b	1.77 ± 0.09 e
Mn + SA2	688.34 ± 19.45 b	277.88 ± 18.81 de	0.86 ± 0.02 b	0.35 ± 0.02 de	2.48 ± 0.07 b
Mn + CA	8.94 ± 1.22 e	477.94 ± 12.30 a	0.01 ± 0.01 e	0.60 ± 0.02 a	0.02 ± 0.01 h
Mn + CA + ALA1	819.34 ± 39.49 a	348.64 ± 7.40 bc	1.02 ± 0.05 a	0.44 ± 0.01 bc	2.35 ± 0.11 bc
Mn + CA + ALA2	603.73 ± 30.06 bc	331.41 ± 6.76 c	0.75 ± 0.03 bc	0.41 ± 0.01 c	1.82 ± 0.09 de
Mn + CA + SA1	624.59 ± 48.26 bc	298.57 ± 3.26 cd	0.78 ± 0.06 bc	0.37 ± 0.01 cd	2.09 ± 0.16 cd
Mn + CA + SA2	602.74 ± 21.84 bc	216.54 ± 6.57 f	0.75 ± 0.03 bc	0.27 ± 0.01 f	2.78 ± 0.10 a

Note: Each value is the mean of three individual replicates. Means followed by the same letter are not significantly different at *p* < 0.05. CA means 20 mmol kg^−1^ citric acid treatment. ALA1/ALA2 indicates spraying with 10/20 mg L^−1^ 5-aminolevulinic acid. SA1/SA2 indicates spraying with 50/100 mg L^−1^ salicylic acid. Mn means that plants exposed to 0.8 g kg^−1^ of manganese for 7 days; Mn + CA indicates plants treated with CA for 7 days after Mn treatment. Mn + CA + ALA/SA treatment indicates plants treated with ALA/SA for one week after Mn + CA treatment.

**Table 4 antioxidants-12-00580-t004:** Effects of ALA/SA combined with citric acid (CA) on manganese (Mn) bioaccumulation quantity (BCQ) and remove efficiency (RE) in sunflower under Mn treatment.

Treatments	Root Mn BCQ (μg)	Shoot Mn BCQ (μg)	Total Mn BCQ (μg)	Remove Efficiency (%)
Mn	29.40 ± 2.62 e	203.85 ± 16.70 f	233.25 ± 19.32 f	0.19 ± 0.016 e
Mn+ALA1	54.97 ± 4.80 c	260.37 ± 19.00 e	315.34 ± 23.80 e	0.26 ± 0.019 d
Mn+ALA2	42.82 ± 0.65 d	380.72 ± 23.03 abc	423.54 ± 23.68 bc	0.35 ± 0.020 b
Mn+SA1	64.72 ± 3.02 b	428.77 ± 22.71 a	493.49 ± 25.73 a	0.41 ± 0.019 a
Mn+SA2	33.98 ± 2.30 e	361.93 ± 10.22 bcd	395.91 ± 12.52 cd	0.33 ± 0.008 b
Mn + CA	83.45 ± 2.15 a	6.96 ± 0.95 g	90.41 ± 3.1 g	0.08 ± 0.001 f
Mn+CA+ALA1	42.53 ± 0.90 d	421.63 ± 20.32 ab	464.16 ± 21.22 ab	0.39 ± 0.017 ab
Mn+CA+ALA2	47.62 ± 0.97 d	354.27 ± 17.64 cd	401.89 ± 18.61 bc	0.33 ± 0.015 bc
Mn+CA+SA1	42.54 ± 0.46 d	392.99 ± 30.36 abc	435.53 ± 30.82 abc	0.36 ± 0.026 ab
Mn+CA+SA2	18.56 ± 0.56 f	320.23 ± 11.60 d	338.79 ± 12.16 de	0.28 ± 0.010 cd

Note: Each value is the mean of three individual replicates. Means followed by the same letter are not significantly different at *p* < 0.05. CA means 20 mmol kg^−1^ citric acid treatment. ALA1/ALA2 indicates spraying with 10/20 mg L^−1^ 5-aminolevulinic acid. SA1/SA2 indicates spraying with 50/100 mg L^−1^ salicylic acid. Mn means that plants exposed to 0.8 g kg^−1^ of manganese for 7 days; Mn + CA indicates plants treated with CA for 7 days after Mn treatment. Mn + CA + ALA/SA treatment indicates plants treated with ALA/SA for one week after Mn + CA treatment.

## Data Availability

Data are contained within the article and Appendix A.

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
