# Peer review of "Promotive Role of 5-Aminolevulinic Acid or Salicylic Acid Combined with Citric Acid on Sunflower Growth by Regulating Manganese Absorption"

_antioxidants, 2023, doi:10.3390/antiox12030580_

Round 1
Reviewer 1 Report
The uptake of the necessary amount of manganese is very important for plants, since it is a vital micronutrient, but it may have toxic effects in greater concentration. Therefore the better understanding of its uptake, transport and protective processes are very important. The authors studied the influence of various compounds on these processes. The abstract should better explain the obtained relationships instead the listing of the measured parameters. Figure 7 consists of many graphs and it should be replaced by a heatmap for better overview. It is not clear, why it does not contain all treatments shown in Figure 6. In section 3.7 a correlation between the concentrations of the various elements is mentioned, but their measurements is not described in the method and results sections. A model figure summarising the similarities and differences between the effects of the various treatments on the studied parameters would help to better understand the discussion of the results. Certain sentences are not clear, and I found mistakes in the English and typing, too.
Minor remarks
1. l. 28: Please, rewrite the first part of the sentence!
2. l. 71: Please, check the sentence!
3. l. 79: Was it irrigation?
4. l. 114: …determined in a reaction mixture…
5. l. 120: GR?
6. l. 181: Please, explain the abbreviations!
7. Fig. 2: The explanation of columns’ filling is too small. It would be enough one explanation below all graphs. Then the graphs could be also larger. Fig. 1 and 3 could be similar.
8. l. 343: …mechanisms-related proteins…
9. 3.7 section: Correlations are mentioned only for few parameters.
10. l. 442: …activity of antioxidant…
11. l. 493: …gene expression…
12. l. 500-513: not clear
Author Response
Dear reviewer 1,
Many thanks for your efforts in reviewing our manuscript and giving us the opportunities to improve our manuscript. We have read and considered all the comments carefully and revised our manuscript accordingly.
Please find the response to all your comments point by point in the attached file.
Thank you very much.
Best regards

Reviewer 2 Report
The manuscript entitled ’Promotive role of ALA/SA combined with citric acid on sunflower growth by regulating manganese absorption (no.: 2192864)’ is about a study on the effects of citric acid, 5-aminolevulinic acid, and salicylic acid on manganese (Mn) absorption in sunflower plants. The study aims to explore the synergistic effects of these compounds on Mn absorption and the plant's antioxidant defense system, ROS levels, cellular changes, and stress-related gene expression. The study is a good and interesting work which belongs to chemical hardening of sunflower plants.
The results show that citric acid (CA) activates Mn in soil and increases its absorption in sunflower roots, but does not improve its translocation efficiency. On the other hand, 5-aminolevulinic acid and salicylic acid (ALA) regulate the translocation efficiency and promote the transportation of Mn from roots to shoots. The study provides insight into the mechanism of Mn translocation and offers an ideal method for regulating Mn absorption in soil with Mn deficit.
The key findings of the study: CA increased sunflower growth, absorption of Mn, and antioxidant activity under 0.8 g kg-1 Mn treatment. The effects of the combination of CA and ALA/SA on Mn phytoextraction in sunflowers were complex. CA increased Mn bioavailability in soil, but inhibited transport of Mn from root to shoot. ALA/SA improved Mn absorption by shoots but reduced it in roots under Mn + CA stress. ALA/SA increased root Mn concentration, Mn bioaccumulation and root damage under Mn treatment.
Before publish, I have got some minor remarks:
- - Remove double dot at row 205 or remove ’L’ from the end of taxon name.
- - Once Fig. 6 and Fig. 7 present the same results, I recommend, that kepp Fig. 6. as main figure and present Fig. 7. as supplementary material. Figure 6. is better to understand the gene expression changes and if readers want to see numeric data and statistics, they can open the supplementary file.
It is really advantageous to present electron micrographs from leaf mesophyll cells under the various treatments.
In my opinion, the text is well-written and clear, the figures are ready in their present form. Authors’ work help to get a deeper insight into the action of Mn absorption in sunflower and maybe in other plants. I recommend this work to publish it in the present form.
Author Response
Dear reviewer 2,
Many thanks for your efforts in reviewing our manuscript and sending us constructive comments to improve the quality of our manuscript. We have read and considered all the comments carefully and revised our manuscript accordingly.
We have response your comment point by point as followings:
Before publish, I have got some minor remarks:
- Remove double dot at row 205 or remove ’L’ from the end of taxon name.
Response: Thanks for your kind comment. We have removed “L.” from the end of taxon name.
- Once Fig. 6 and Fig. 7 present the same results, I recommend, that kept Fig. 6. as main figure and present Fig. 7. as supplementary material. Figure 6. is better to understand the gene expression changes and if readers want to see numeric data and statistics, they can open the supplementary file.
Response: Thanks for your kind suggestion. We have removed Figure 7 from the manuscript to the supplementary material Figure S2. If readers want to see numeric data and statistics, they can open the supplementary Figure S2 and Table S7 files.
It is really advantageous to present electron micrographs from leaf mesophyll cells under the various treatments.
Response: Thanks for your positive comment.
In my opinion, the text is well-written and clear, the figures are ready in their present form. Authors’ work help to get a deeper insight into the action of Mn absorption in sunflower and maybe in other plants. I recommend this work to publish it in the present form.
Response: Thanks for your positive comments
Thanks again!
Round 2
Reviewer 1 Report
The authors made the suggested changes. MInor spell check is necessary, For exmple: l. 317: We should be changed to we.
Author Response
Dear reviewer,
Many thanks for giving us the opportunity to improve our manuscript (antioxidants-2192864). We have carefully read this manuscript and revised accordingly.
The revised parts were marked with Track Changes function in our revised manuscript.
Your mentioned "We" has been changed into "we" in L 523.
Thanks again!
Best regards,